# Does the Dizziness Handicap Inventory—Children and Adolescents (DHI-CA) Demonstrate Properties to Support Clinical Application in the Post-Concussion Population: A Rasch Analysis

**DOI:** 10.3390/children10091428

**Published:** 2023-08-22

**Authors:** Devashish Tiwari, Perman Gochyyev

**Affiliations:** MGH Institute of Health Professions, Boston, MA 02129, USA; pgochyyev@mgh.harvard.edu

**Keywords:** children, concussion, self-report measure, dizziness, disability

## Abstract

The purpose of this cross-sectional validation study was to evaluate the clinical utility of the DHI-CA by (1) examining its dimensionality using exploratory factor analysis (EFA) and (2) calibrating DHI-CA items (using the multidimensional Rasch model) to obtain item difficulty levels. A retrospective chart review was conducted for 132 patients between the ages of 8 and 18 years (mean age = 15.3 ± 2.1 years) from a multidisciplinary post-concussion management tertiary center. Data were extracted on age, sex, and DHI-CA. EFA revealed that 12 out of 25 items did not fit in the subscale that they were originally described under, indicating poor dimensionality. Calibration of items on the Wright Maps revealed that 50% of the items pooled in the lower difficulty level, indicating a potential ceiling effect. Corrected item–rest correlations for the physical, emotional, walking/mobility, and community participation ranged from 0.44–0.66, 0.27–0.61, 0.54–0.57, and 0.32–0.69 (*p* < 0.001), respectively. The clinical utility of the DHI-CA was found to be questionable due to the presence of double-barreled items and the ceiling effect. Clinicians must supplement data from the DHI-CA with other measures and patient interviews to make informed clinical decisions specific to the post-concussion population until new, robust, and valid measures are developed.

## 1. Introduction

The incidence of concussion in children has risen by 60% since 2007, presenting a cause of significant concern, primarily due to the associated lasting symptoms such as dizziness [1]. The occurrence of dizziness post-concussion is alarmingly high (up to 80%) among children [2]. Dizziness post-concussion can originate from multiple factors, including peripheral vestibular dysfunction, cervical spine, and/or cardiovascular system dysfunction [3]. Dizziness is often associated with other symptoms, including poor coordination, light-headedness, poor balance, and frequent falls, which can significantly impair community participation [4]. More concerning is that, in children, dizziness negatively affects psychological state, academic performance, participation in sports, and activities of daily living, thereby significantly impacting the overall quality of life [5]. Children who have dizziness are 6.6 times more likely to have an intellectual disability and 2.46 times more likely to use special education services when compared to children without dizziness [6]. Early identification and assessment of dizziness is, therefore, crucial to initiate timely referrals to rehabilitation intervention to achieve better outcomes [7].

Patient-reported outcome measures are often utilized in clinics to identify the patient’s perception of activity-specific functional limitations [8]. Identifying these areas assist clinicians in designing targeted intervention strategies and evaluating the effectiveness of targeted rehabilitation interventions. The Dizziness Handicap Inventory—Children and Adolescents (DHI-CA) is a new patient-reported measure to assess the impact of dizziness. The DHI-CA was constructed from the Brazilian-Portuguese version of the DHI for adults, and it underwent semantic adaptation for children between 6–14 years [9]. Since DHI-CA is the only patient-reported outcome measure currently available to assess the impact of dizziness in children post-concussion, examining its clinical utility is of crucial importance. 

Previous studies highlighted notable structural limitations in the DHI-CA, including questionable construct validity and high item redundancy, and recommended further investigation of its factorial structure using Item Response Theory (IRT) methods [10]. IRT is a statistical framework that allows the analysis of measurement properties of the items and the underlying factorial structure measured by the items. IRT models, and the Rasch model in particular, estimate the item difficulty (that allows ordering the items from least to most difficult to perform) and allows evaluating the match between items and patient performances (i.e., the extent of the appropriateness of item difficulty) and person (e.g., patient) separation (the ability of the items to differentiate between distinct strata of functioning) [11]. Using the Rasch model for calibrating obtained measures allows converting the arbitrary intervals between the responses recorded on a Likert scale to empirically obtained distances, hence changing the scores on the ordinal scale to interval-level scores [12]. The Rasch model belongs to the IRT family of models and has been utilized to improve patient-reported outcome measures due to its rigor and interpretational advantages. Specifically, one of the advantages of the Rasch model is its rigorous examination of the fit of each item to the model, which is particularly helpful for developing a new measure. 

This study aimed to examine the measurement properties of the DHI-CA in the post-concussion population using EFA, followed by the evaluation of the items using the multidimensional Rasch modeling approach. Specifically, we aim to evaluate the clinical utility of the DHI-CA by (1) examining the dimensionality of the DHI-CA using EFA and (2) calibrating DHI-CA items (using the multidimensional Rasch model) to obtain item difficulty levels. The results of this study will provide important information regarding the robustness and applicability of this measure that will guide clinicians in making informed clinical decisions when evaluating the impact of dizziness for children post-concussion.

## 2. Materials and Methods

### 2.1. Study Design and Participants

This was a cross-sectional retrospective study. The data was collected from the electronic medical records of a tertiary center specializing in comprehensive interdisciplinary management of patients’ post-concussion. The institutional review board of the University of Michigan-Flint approved this study (HUM00126048). This study utilized the data of 132 children and adolescents between the ages of 8 and 18 years who were referred to vestibular physical therapy post-concussion. 

### 2.2. Measures

This study included data on age (in years), sex, DHI-CA scores at initial evaluation, and discharge from physical therapy. The DHI-CA is comprised of twenty-five items that are categorized under three sub-scales (i.e., a priori specified dimensions), which are (1) physical (7 items), (2) emotional, and (3) functional (9 items each). Each item is scored on a 3-point ordinal scale (no dizziness, sometimes, and always). For simplicity in the use and interpretation, the three levels were scored as 0, 2, and 4, resulting in the DHI-CA scores ranging from 0–100, with higher scores indicating higher perceived disability [9]. Evidence of the DHI-CA measure’s construct validity and reliability has been reported in the literature [9,10]. 

### 2.3. Statistical Analysis

DHI-CA scores from the initial evaluation were used in this study. EFA was performed with quartimin rotation to examine the factorial structure of the DHI-CA. Factors with eigenvalues > 1 were reviewed for retention in the model [13]. Bartlett’s test was performed to examine the sphericity and determine the appropriateness of this data for the EFA. Moreover, the Kaiser–Meyer–Olkin (KMO) test was conducted to evaluate sample adequacy [14,15]. Based on results obtained from the EFA, the new factorial structure was cross-examined with the factorial structure reported in the original validation study. 

Following this, a multidimensional (four-dimensional) ordinal Rasch modeling approach was used. The ordinal Rasch model, also known as the partial credit model, was used due to its ability to handle categorical items with more than two scoring levels [16]. The model estimation was performed using the marginal maximum likelihood method [17]. Additionally, item-specific weighted mean-square (infit) statistics were utilized to determine the appropriateness of the multidimensional Rasch model and evaluate its fit [11].

Finally, the Wright Map, which represents the estimated distribution of respondents across each factor (shown on the left, with the highest scoring respondents located at the top), was obtained. The right side of the map shows the estimated distribution of item difficulties, with the most “difficult” activity at the top. Mapping the items according to the difficulty level provides a robust methodological perspective for data collection when making informed clinical decisions.

## 3. Results

### 3.1. Demographics

The average age of the participants was 15.3 ± 2.1 years (40.2% males). Nineteen percent of participants sustained a concussion after contacting the playing surface, 34% of the injuries resulted from contact with another player, and 22% of sustained injuries from contacting sporting equipment. Soccer (16.7%), basketball (14.4%), and football (12.1%) were the top three sports that resulted in children sustaining concussions. The mechanism of injury was not sports-related for 23% of participants. 

### 3.2. Factor Analysis

A significant Bartlett’s test (*p* < 0.001) indicated that it was appropriate to perform factor analysis. The KMO value (KMO = 0.845) indicated that the sample was adequate to perform factor analysis. A four-factor solution was endorsed (based on the rule of eigenvalue > 1). We found that factor 1 (physical) explained 23.8% of the total variance, whereas factors 2 (emotional), 3 (walking and mobility), and 4 (community participation) explained 21.1%, 18.6%, and 10.5% of the total variance, respectively. 

The results from EFA indicated that 12 out of 25 items did not fit in the subscale that they originally were described under (Table 1). Based on factor loading values, a revised factor structure was created by re-allocating all the items from the functional subscale to physical (3 items), walking and mobility (3 items), and community participation (3 items) factors (Table 1). The measurement properties of the DHI-CA are described as follows. Based on the multidimensional Rasch model, the four factors demonstrated a high degree of association with each other (Table 2). 

### 3.3. Item Difficulty

Figure 1 represents item difficulty distribution, whereas item fit measures and item–rest correlations are reported in Table 3. Overall, item 9 (Because of the dizziness, are you afraid to leave the house?) demonstrated the lowest difficulty level, followed by item 16 (Because of the dizziness, is it too difficult for you to walk about alone?). On the other hand, item 11 (Do fast movements of the head worsen your dizziness?) demonstrated the highest difficulty, followed by item 8 (Do games, sports, riding a bicycle, riding on merry-go-rounds worsen the dizziness) and item 18 (Because of dizziness, do you have difficulty with concentrating on your school activities) (Table 3). 

Corrected item–rest correlations for the physical, emotional, walking/mobility, and community participation ranged from 0.44–0.66, 0.27–0.61, 0.54–0.57, and 0.32–0.69, respectively. The lowest item–rest correlation was observed for item 21 (Because of the dizziness, do you feel harmed (a) in comparison with your colleagues?/Companions?), whereas the highest item–rest correlation was observed for item 18 (Because of the dizziness, do you have difficulty with concentrating on your school activities?).

### 3.4. Reliability

Based on the chi-square test to compare the models, the four-dimensional model has a statistically significantly better (at *p* < 0.001 level) fit compared to the unidimensional Rasch model. The Akaike Information Criterion (AIC) and Bayesian Information Criterion (BIC) also supported the model. Overall reliability and between-subscale scores were comparable between the two models of the DHI-CA except for the “physical” subscale, which demonstrated better internal consistency with the four-factor model (EAP reliability = 0.91 vs. 0.86). 

The four-factor model demonstrated strong internal consistency, with Cronbach’s α values ranging between 0.86–0.89 for the individual subscales (Table 4). Individually, each subscale demonstrated acceptable internal consistency, with Cronbach’s α estimated at 0.91 for the overall measure and expected-a-posterior (EAP) reliabilities estimated at 0.87, 0.78, 0.83, and 0.83 for physical, emotional, walking/mobility, and community participation, respectively. This suggests four-factor model demonstrated internal consistency better than the previously reported model [9]. 

## 4. Discussion

The present study aimed to investigate the clinical utility of the DHI-CA by examining its underlying factorial structure and measurement properties in the post-concussion population using multidimensional Rasch models. While DHI-CA is the only available measure to assess dizziness in children post-concussion, our results indicate critical concerns in measurement properties, making its clinical applicability questionable. 

Our analysis identified a different factor structure than what was originally reported for the DHI-CA. The “functional” subscale originally reported items representing both mobility and community participation—two notably different constructs. Items addressing activities like studies, participation in school activities, participating in birthday parties, playing video games, and staying away from school are very specific to community participation (home and school), while items such as the inability to walk in the dark, walking on a sidewalk/ground full of holes, staying away from high places, and being afraid to leave the house represent individual activities covered under the construct of mobility. 

Another area of concern noted was around item difficulty and ceiling effect. The Wright Maps revealed that 50% of the items were pooled in the lower difficulty level (below two standard deviations from the midpoint on the Wright Map), indicating a potential ceiling effect for the current sample. Items that demonstrated that the lowest difficulty level (afraid to leave the house alone, feeling sad, staying away from school, and staying away from high places) might not present an adequate level of challenge to the participants in the current sample. Hence, the development of a new measure with the equivalent distribution of items across the difficulty level is warranted.

There were several double-barreled items (single item containing different constructs) in the measure (e.g., Do games, sports, riding a bicycle, riding on roundabouts/merry-go-rounds worsen the dizziness; Because of the dizziness, do you find it difficult to jump, run, play ball games, ride a bicycle?). It is noteworthy that riding on merry-go-rounds and riding a bicycle present different intensities of challenge to the vestibular system. Similarly, playing video games and watching movies are activities that do not require mobility as compared to attending birthdays and parties. Hence, it is recommended that these constructs should not be included in a single item so a better distinction between activity limitation and participation restriction can be obtained. The presence of double-barreled items in a measure can potentially increase the cognitive load for the respondents and may present inaccurate data, thereby affecting data quality [18]. It is likely that while answering a double-barreled question, the respondents may disregard one aspect of the question or produce a collective response for both dimensions of the question [18].

Multiple items in the DHI-CA contain negative formulation (for example, Because of the dizziness, do you have difficulty with concentrating on your school activities; Do you stay away from birthdays, parties, movies, video game arcades because of the dizziness; Because of the dizziness, is it too difficult for you to walk about alone?). Unlike self-report measures used for adults, where an equal balance of positively and negatively phrased items is recommended, negatively phrased items must not be included in a self-report measure for children as it may degrade the data quality [19]. Finally, the expert pool in creating the original DHI-CA was limited to speech-language pathologists [9]. Since dizziness affects multiple aspects of functions, it is important to have multidisciplinary team members review the measure and provide constructive inputs [20,21]. There were some limitations in this study. This study included only the post-concussion population. It is possible that the DHI-CA may have better clinical utility in populations who experience dizziness from other causes (motion sickness, hearing impairments, etc.). Future studies may explore the model fit separately by specific medical diagnosis.

## 5. Conclusions

The Rasch analysis indicated several structural limitations in the DHI-CA. The presence of double-barreled items and a potential ceiling effect necessitate caution while interpreting the results of the DHI-CA in children post-concussion. Therefore, until new, robust measures are developed, clinicians must supplement data from the DHI-CA with other measures and patient interviews to make informed clinical decisions specific to the post-concussion population. Developing a new patient-reported outcome measure with improved measurement properties specific to the post-concussion population is warranted. 

## Figures and Tables

**Figure 1 children-10-01428-f001:**
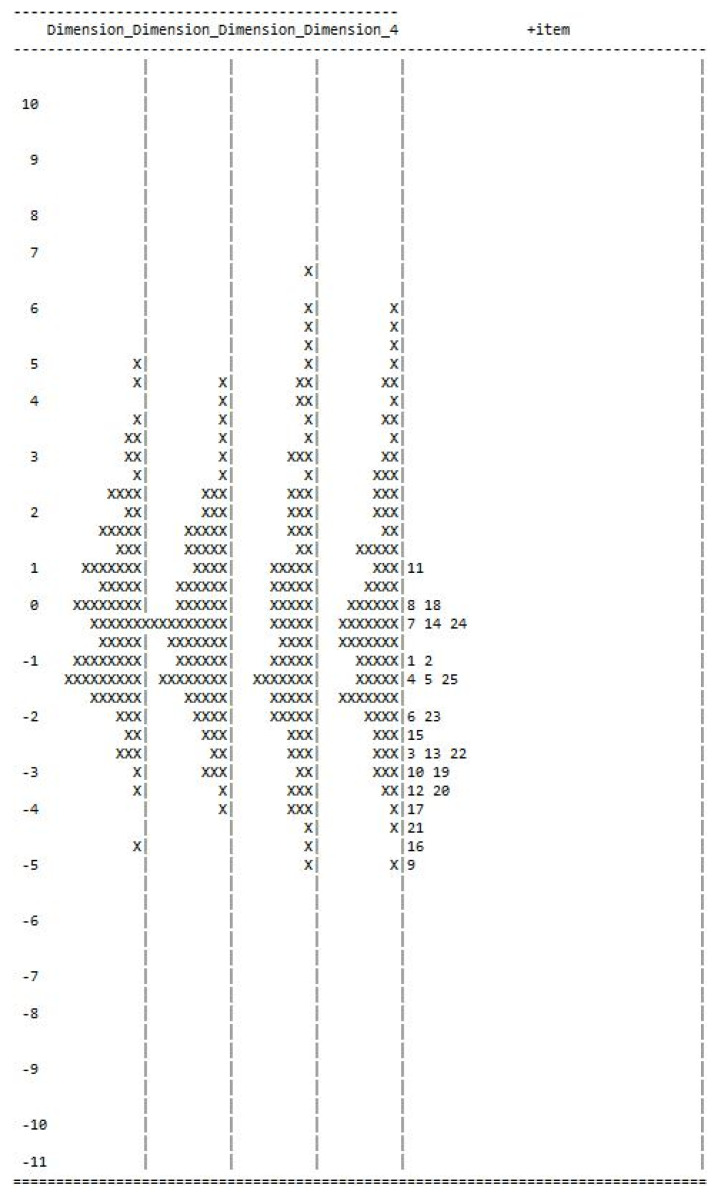
Wright Maps for item difficulty distribution. Note: Items towards the bottom represent lower difficulty levels.

**Table 1 children-10-01428-t001:** Factor structure comparison by exploratory factor analysis.

Item Number	Item Description	Original Factor	New Factor	Loading Value
1	Does lifting your head up worsen dizziness?	Physical	Physical	0.667707
2	Because of the dizziness, do you feel frustrated (a)?	Emotional	Emotional	0.471055
3 *	Because of the dizziness, do you stay away from school?	Functional	Community participation	0.654678
4	Does walking around the supermarket looking at the shelves worsen the dizziness?	Physical	Physical	0.471355
5 *	Because of the dizziness, do you have difficulty with getting up from the bed?	Functional	Physical	0.615827
6 *	Do you stay away from birthdays, parties, movies, video game arcades because of the dizziness.	Functional	Community participation	0.574299
7 *	Because of the dizziness, do you have difficulty with reading?	Functional	Physical	0.583219
8	Do games, sports, riding a bicycle, riding on roundabouts/merry-go-rounds worsen the dizziness?	Physical	Physical	0.666689
9 *	Because of the dizziness, are you afraid to leave the house?	Emotional	Walking/mobility	0.652956
10	Because of the dizziness, do you feel ashamed (a) in front of others?	Emotional	Emotional	0.739026
11	Do fast movements of the head worsen your dizziness?	Physical	Physical	0.722264
12 *	Because of the dizziness, do you stay away high places?	Functional	Walking/mobility	0.546666
13	If you turn in bed while you are lying down (a) does it worsen your dizziness?	Physical	Physical	0.640493
14 *	Because of the dizziness, do you find it difficult to jump, run, play ball games, ride a bicycle?	Functional	Physical	0.640339
15	Because of the dizziness, are you afraid that people will think you are not well?	Emotional	Emotional	0.546115
16 *	Because of the dizziness, it too difficult for you to walk about alone (a)?	Functional	Walking/mobility	0.828326
17 *	Does walking on the sidewalk, passing or going over a ground full of holes worsen the dizziness	Physical	Walking/mobility	0.470037
18 *	Because of the dizziness, do you have difficulty with concentrating on your school activities?	Emotional	Community participation	0.450484
19 *	Because of the dizziness, are you unable to walk about in the dark?	Functional	Walking/mobility	0.426231
20	Because of the dizziness, are you afraid to stay at home alone?	Emotional	Emotional	0.410947
21	Because of the dizziness, do you feel harmed (a) in comparison with your colleagues?/Companions?	Emotional	Emotional	0.302822
22	Because of the dizziness, do you quarrel with your friends, companions or persons in your family?	Emotional	Emotional	0.579006
23	Because of the dizziness, do you feel sad, without wanting to do anything?	Emotional	Emotional	0.715102
24 *	Does your dizziness hamper, interfere in your studies?	Functional	Community participation	0.408427
25	If you lower your head or body, does the dizziness worsen?	Physical	Physical	0.594282

Notes: * = Items classified under different categories from the original distribution.

**Table 2 children-10-01428-t002:** Correlation matrix (and variances in the diagonals) from the 4-factor CFA model.

	Physical *	Emotional *	Walking/Mobility *	Community Participation *
Physical	3.352 (0.413)			
Emotional	0.661	3.838 (0.472)		
Walking/mobility	0.823	0.870	7.713 (0.949)	
Community participation	0.741	0.770	0.789	5.185 (0.683)

Note: Standard errors are shown in parenthesis; * = *p* < 0.001.

**Table 3 children-10-01428-t003:** Corrected item–rest correlation values for individual items.

Item Description	Item–Rest Correlation	Item Difficulty (SE)	Item Fit. (Unweighted MNSQ)	Item Fit. (Weighted MNSQ)
**Physical**				
Does lifting your head up worsen dizziness?	0.52 *	1.002 (0.183)	1.00	1.03
Does walking around the supermarket looking at the shelves worsen the dizziness?	0.60 *	1.423 (0.166)	0.95	1.06
Because of the dizziness, do you have difficulty with getting up from the bed?	0.57 *	1.277 (0.165)	0.99	1.05
Because of the dizziness, do you have difficulty with reading?	0.66 *	0.175 (0.157)	0.94	0.96
Do games, sports, riding a bicycle, riding on roundabouts/merry-go-rounds worsen the dizziness	0.56 *	−0.19 (0.170)	0.96	0.97
Do fast movements of the head worsen your dizziness?	0.53 *	−0.830 (0.170)	0.90	0.93
If you turn in bed while you are lying down (a) does it worsen your dizziness?	0.44 *	2.894 (0.206)	0.79	1.01
Because of the dizziness, do you find it difficult to jump, run, play ball games, ride a bicycle?	0.61 *	0.084 (0.164)	0.91	0.93
If you lower your head or body, does the dizziness worsen?	0.54 *	1.246 (0.166)	1.02	1.12
**Emotional**				
Because of the dizziness, do you feel frustrated (a)?	0.53 *	1.082 (0.168)	0.97	1.08
Because of the dizziness, do you feel ashamed (a) in front of others?	0.41 *	3.408 (0.247)	0.99	0.79
Because of the dizziness, are you afraid that people will think you are not well?	0.57 *	2.372 (0.191)	0.72	0.89
Because of the dizziness, are you unable to walk about in the dark?	0.61 *	3.325 (0.221)	1.05	1.09
Because of the dizziness, are you afraid to stay at home alone?	0.40 *	3.639 (0.267)	1.11	1.11
Because of the dizziness, do you feel harmed (a) in comparison with your colleagues?/Companions?	0.27 **	4.329 (0.299)	1.25	1.26
Because of the dizziness, do you quarrel with your friends, companions or persons in your family?	0.47 *	2.894 (0.217)	0.86	1.06
Because of the dizziness, do you feel sad, without wanting to do anything?	0.54 *	2.252 (0.185)	0.89	1.03
**Walking/Mobility**				
Because of the dizziness, are you afraid to leave the house?	0.56 *	5.045 (0.298)	0.43	0.89
Because of the dizziness, do you stay away high places?	0.57 *	3.528 (0.241)	1.02	1.39
Because of the dizziness, it too difficult for you to walk about alone (a)?	0.54 *	4.773 (0.280)	0.70	0.90
Does walking on the sidewalk, passing or going over a ground full of holes worsen the dizziness	0.56 *	4.030 (0.251)	0.82	1.04
**Community participation**				
Because of the dizziness, do you stay away from school?	0.32 *	2.885 (0.209)	1.05	1.29
Do you stay away from birthdays, parties, movies, video game arcades because of the dizziness.	0.53 *	2.034 (0.184)	1.10	1.04
Because of the dizziness, do you have difficulty with concentrating on your school activities?	0.69 *	−0.205 (0.173)	0.99	0.88
Does your dizziness hamper, interfere in your studies?	0.65 *	0.216 (0.173)	0.89	0.95

Note: * = (*p* < 0.001), ** = (*p* = 0.001).

**Table 4 children-10-01428-t004:** Model fit and reliability estimates in the 4-dimensional model.

Statistics	4-Dimensional Model
Sample size	132
Number of items in the calibration	25
Missing data	1.21%
Model	PCM
Weighted fit MNSQ > 1.33, T sig. at 0.05 (Item Parms)	1 item
Reliability estimates	
EAP reliability	
Physical	0.878
Emotional	0.789
Walking/mobility	0.830
Community participation	0.832
Cronbach’s Alpha	
Overall	0.91
Physical	0.89
Emotional	0.86
Walking/mobility	0.88
Community participation	0.90
Deviance	4111.51
Number of estimated parameters	60
AIC	4194.11
BIC	4400.47

Notes: AIC = Akaike Information Criterion, BIC = Bayesian Information Criterion.

## Data Availability

Data are unavailable due to privacy.

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
