# Peer review of "Does the Dizziness Handicap Inventory—Children and Adolescents (DHI-CA) Demonstrate Properties to Support Clinical Application in the Post-Concussion Population: A Rasch Analysis"

_children, 2023, doi:10.3390/children10091428_

Round 1
Reviewer 1 Report
The authors present a manuscript describing a cross sectional validation study examining the measurement properties of the DHI_CA in the post-concussion population using exploratory factor analysis followed by evaluation of items using a multidimensional RASCH modeling approach. Overall, the article is well conceived, and the statistical analysis done is explained well. Suggestions below are to improve the clarity and scientific aspects of the manuscript:
-Table and Figure formats in the manuscript are hard to follow. It may be due to manuscript formatting but please more clearly present the tables and figures and align them with the specific results section.
-Citations
the current manuscript only includes citations 1-12. In the discussion there is reference to citations 18 and 19 but they are not included in the list. Please update the reference list to include all citations.
-Discussion
page 8 lines 184-190. The content of this paragraph is not clear. How did you determine that the items pooled were in the "lower difficulty level"?
page 8 Lines 197-201- It is unclear what this means because the citation for #18 in not listed in the references.
page 8 lines 204 210. It is not clear what this means because the citation for #19 is not included in the reference list.
Double check English grammar.
Author Response
Thank you for your insightful feedback. We appreciate it. Please find attached our detailed response.

Reviewer 2 Report
Authors have written this article well. They may consider few suggestions to help the broader audience and provide transparency. Reflecting on the impact this may have clinically, wording as "must" is relevant when the research is compelling, which in this case the case is yet to be made. Transparency in Methods may help.
Redundancy in wording: Line 8 study, as well as Line 19
Methods with Line ref:
1) 43-50:
Could you provide more background on the DHI-CA and its importance in the context of this study? Examples that indicate validity?
What are these structural limitations explicitly so we can see clearly what about IRT improves.
61-65:
What are the expected contributions / implications of the research?
Line 88: Elaborate and explain why this is significant.
Discussion:
How do the limitations affect our ability to generalise and what directions do you suggest for future research moving forward?
Good
Author Response

(The authors gave the same response as above.)

Reviewer 3 Report
Thanks for the opportunity to review this study, which presents a very important topic for child otoneurology, I congratulate the authors for the construction of this study. My considerations are detailed below:
Abstract:
- For a better understanding of the readers, I suggest that the authors include the subtopics in the abstract (introduction, objectives, methods, results and conclusion).
- I suggest that authors put the values of correlations and p-values right after the outcome that was analyzed.
- The abstract must be rewritten so that the conclusions respond to the objective.
Introduction:
- The introduction presents some paragraphs that do not demonstrate fluidity in the text, that is, they end a paragraph talking about a topic, and start the next one on a completely different topic. Example: “ ... Dizziness lasting more than three weeks is an independent predictor of prolonged recovery time[5]. In children, post-concussion dizziness negatively affects academic performance, participation in sports, activities of daily living,” The authors should improve the fluidity of the introduction text.
- I believe that the authors could describe more about the impact of dizziness on the aforementioned aspects of the life of the child and adolescent.
Methods:
- The authors use the term “gender” to refer to the sex of the child, which is not the same thing. Sex is what you are born with, and gender is what you understand yourself as a person, so I can have a certain sex and identify/recognize myself with it or not, that is gender. I suggest that authors be careful when using the term gender.
Results:
- In the demographic data, do the authors have information on which sports were the ones in which 19% of children suffered concussions? If so, I suggest including it, because these data may be important for other studies and this increases the chances of this article being cited by others.
- In some analyses, the p-values only appear in the text of the results, why is this p-value not in the tables and figures?
Discussion:
- One question is whether the DHI-CA is a questionable instrument to be applied to children with post-concussion dizziness, or to other children as well? This needs to be better specified by the authors. Because, for example, deaf children, children with motion sickness, have dizziness, is the application of DHI-CA not recommended for these children?
References:
- Only 12 references is too little to demonstrate the scientific robustness of a topic in a scientific article. I suggest authors to include more references, to demonstrate the importance of the study and the topic and to improve the scientific robustness of this study.
- Only two references from the last 5 years. Where is the recent evidence on the topic?
- Reference 9 has the wrong name of the journal quoted.
Author Response

(The authors gave the same response as above.)

Round 2
Reviewer 1 Report
The authors have addressed all comments- thank you.
Reviewer 3 Report
Congratulations to the authors, they did a good job. I believe that now the article has a clearer and more understandable text for the readers.